# Smart Anticorrosion Coatings Based on Poly(phenylene methylene): An Assessment of the Intrinsic Self-Healing Behavior of the Copolymer

**DOI:** 10.3390/polym14173457

**Published:** 2022-08-24

**Authors:** Marco F. D’Elia, Mirko Magni, Thomas Romanò, Stefano P. M. Trasatti, Markus Niederberger, Walter R. Caseri

**Affiliations:** 1Laboratory for Multifunctional Materials, Department of Materials, ETH Zürich, 8093 Zürich, Switzerland; 2Department of Environmental Science and Policy, Universitá degli Studi di Milano, 20133 Milan, Italy

**Keywords:** organic coating, poly(phenylene methylene), self-healing, corrosion protection, electrochemical impedance spectroscopy

## Abstract

Poly(phenylene methylene) (PPM) is a multifunctional polymer featuring hydrophobicity, high thermal stability, fluorescence and thermoplastic processability. Accordingly, smart corrosion resistant PPM-based coatings (blend and copolymer) were prepared and applied by hot pressing on aluminum alloy AA2024. The corrosion protection properties of the coatings and their dependence on coating thickness were evaluated for both strategies employed. The accelerated cyclic electrochemical technique (ACET), based on a combination of electrochemical impedance spectroscopy (EIS), cathodic polarizations and relaxation steps, was used as the main investigating technique. At the coating thickness of about 50 µm, both blend and copolymer PPM showed effective corrosion protection, as reflected by |Z|_0.01Hz_ of about 10^8^ Ω cm^2^ over all the ACET cycles. In contrast, when the coating thickness was reduced to 30 µm, PPM copolymer showed neatly better corrosion resistance than blended PPM, maintaining |Z|_0.01Hz_ above 10^8^ Ω cm^2^ with respect to values below 10^6^ Ω cm^2^ of the latter. Furthermore, the analysis of many electrochemical key features, in combination with the optical investigation of the coating surface under 254 nm UV light, confirms the intrinsic self-healing ability of the coatings made by PPM copolymer, contrary to the reference specimen (i.e., blend PPM).

## 1. Introduction

Poly(phenylene methylene) (PPM) is a hydrocarbon polymer with chemical formula (C_6_H_4_(CH_2_))_n_, consisting of phenylene rings bridged by a methylene group [1,2]. Therefore, PPM is structurally located between polyethylene and polyphenylene (Figure 1). Importantly, PPM has been found to possess a unique combination of materials properties, namely high thermal stability [3,4,5], hydrophobicity [4,6], resistance towards oxidizing agents [7] and good barrier properties [4,8]. Accordingly, poly(phenylene methylene) was recently discovered as a promising new candidate in the area of corrosion protection [9,10]. In addition, the fluorescence of PPM [11,12] facilitates detection of failures in the coating upon corrosion (as reported in the Results section).

However, coatings made of PPM tend to crack, which precludes their application both in real systems and as reference samples in comparative studies. This drawback can be overcome with the addition of plasticizers to PPM or by using PPM copolymers with alkoxy side chains along the polymer backbone (Figure 1). This can yield continuous films that can be used as anti-corrosion coatings [9,10]. So far, the corrosion resistance of coatings with PPM blended with the plasticizer benzyl butyl phthalate (BBP) or coatings with PPM copolymers containing octyloxy side groups, which act as internal plasticizer, were investigated using the anodic polarization techniques and potentiostatic measurements, providing information only on the barrier-like behavior in a wide potential window and under long anodic polarization. Additionally, the possibility of the self-healing ability of these coatings was preliminary reported [9,10].

In general, among the techniques used to understand the protection ability of organic coatings, electrochemical impedance spectroscopy (EIS) is found to be one of the most useful in studying coating degradation [13,14,15]. In particular, the estimation of water uptake is a key parameter to evaluate the durability of the “barrier effect” of organic coatings, together with the adhesion and the evolution of the physical-chemical properties during ageing [16,17,18]. Water uptake over time can be estimated from the electric capacitance of the coating, in turn disclosed by fitting the EIS spectra through a proper model or equivalent electric circuit. The quantitative relationship between water uptake and coating capacitance (Equation (1)) was developed by Brasher and Kingsbury [19]:(1)$$\phi \left(t\right)=\frac{\ln\left(\frac{C\left(t\right)}{C_{d}}\right)}{\ln\xi_{w}}$$
with ϕ(t) representing the volume fraction of water in the coating, C(t) as the coating capacitance at time t, $C_{d}$ as the coating capacitance of the dry coating and $\xi_{w}$ as the dielectric constant of pure water. This relationship was used for several years to investigate coatings with thicknesses ranging from 1 μm to 100 μm immersed in sea water [16,20,21,22,23]. However, in order to provide faster indication of corrosion processes at the surface and at the interface of coated metallic substrates, Hollaender et al. [24] implemented the classical EIS method combining cathodic polarization stress (direct current: DC) and EIS measurement (alternating current: AC) in order to accelerate the degradation of the coating system. A further development based on the Hollaender method [24,25] is the accelerated cyclic electrochemical technique (ACET), which comprises a potential relaxation step following each cathodic polarization step to allow the formation of a new stable equilibrium (i.e., a new open circuit potential: OCP) before the next EIS measurement is performed. Thus, the ACET procedure (Figure 2a) is based on the repetition of the sequence stress/relaxation/impedance measurement (DC/OCP/AC) for at least six times, or until the coating is totally damaged. A first EIS measurement is carried out to define the starting electrochemical features of the system that are used as reference values for the evaluation of each subsequent EIS spectrum. Then, a cathodic polarization, DC, with potential lower than −1 V vs. the saturated calomel electrode (SCE) is applied to the system in order to give rise to the hydrolysis reaction of water at the metal-coating interface, leading to the formation of H_2_ and OH^−^. The evolution of these two species speeds up the delamination of the coating; thus, the consequent failure of the system is reflected in a change of the impedance spectrum. During the relaxation time, electrochemical corrosion can take place (at a newly generated solution–metal interface), leading to the dissolution of metal at the substrate surface and, possibly, the accumulation of insoluble corrosion products. As a consequence, a variation of the free corrosion potential is observed due to the migration of the formed ions and the reorganization of the coating matrix [26]. In this way, the coating is degraded and loses adherence with the metallic substrate due to the mechanical action of H_2_ evolution, because of the pore opening by the diffusion of electrolyte and corrosion products (Figure 2b).

In the above light, the ACET technique finds wide application in research and in industry to study and evaluate the permeability of coating systems and properties related to the adhesion of the substrate [13,25,26,27]. In this study, a comparison of the corrosion protection properties of PPM-based coatings with external (blend PPM) or internal plasticizers (PPM copolymer) on pretreated AA2024 aluminum alloy is performed by accelerated electrochemical techniques (ACETs). By optimizing the coating thickness, the best coating fabrication strategy is determined. In addition, the self-healing of the PPM copolymer coating is investigated by a combination of electrochemical methods and fluorescence studies using optical microscopy.

## 2. Experimental

### 2.1. Materials

Benzyl chloride stabilized by propylene oxide (99%), tin(IV) chloride (98%), thionyl chloride (99%), sodium sulfate (99%), chloroform (99.8%, amylene stabilized) and tetrahydrofuran (99.9%, LiChrosolv) were purchased from Sigma Aldrich (Buchs, Switzerland): benzyltriethoxysilane (96%) from Fluorochem (Hadfield, UK); 4-hydroxybenzaldehyde (98%) and 1-bromooctane (98%) from abcr (Karlsruhe, Germany); sodium borohydride (98%) from VWR Chemicals BDH (Leuven, Belgium); acetonitrile (99.9%), sodium hydroxide (98.66%) and potassium hydroxide (86%) from Fischer Chemicals (Loughborough, UK); methanol (98%) from Merck (Darmstadt, Germany); and bismuth (III) trifluoromethanesulfonate (98%) from Acros Organics (Geel, Belgium), and benzyl butyl phthalate (98%) from Tokio Chemical Industry (TCI) (Eschborn, Germany). High strength aluminum alloy AA2024 substrates 12 cm in length, 3 cm in width and 4 mm in thickness (4.3–4.5% copper, 1.3–1.5% magnesium, 0.5–0.6% manganese and less than 0.5% of other elements) were provided by Aviometal s.p.a (Varese, Italy).

### 2.2. Preparation of Blended PPM

The synthesis of PPM is described in the literature [11]. As previously reported [9], in order to prepare blends of PPM with 5% mol/mol benzyl butyl phthalate (BBP), 200 mg (0.64 mmol) BBP were added to 2 mL tetrahydrofuran (THF) solution comprising 1.2 g PPM under nitrogen. A slow increase in viscosity was observed upon stirring for 3 h. Subsequently, the temperature of the mixture was risen to 60 °C at reduced pressure (150 mbar). After one hour, the mixture was allowed to adopt ambient temperature. Thereafter, the pressure was reduced to 0.9 mbar for 8 h, resulting in 1.378 g of the PPM–BBP blend.

### 2.3. Synthesis of PPM Copolymer

4-Octyloxybenzyl chloride was synthesized in three steps, as reported previously [10], from 4-hydroxybenzaldehyde (see Appendix A). The synthesis of copolymer of phenylene methylene and 4-octyloxyphenylene methylene was performed according to the literature under the same experimental conditions [10]; however, bismuth (III) trifluoromethanesulfonate (0.32 g, 0.39 mmol) was employed as a less hazardous Lewis acid catalyst instead of tin tetrachloride (see Appendix A). The peak positions in ^1^H and ^13^C NMR data, the molecular weight and the thermal properties agreed with those reported previously [10].

### 2.4. Preparation of Coatings

The AA2024 surfaces were pretreated as described earlier [9,10]. PPM-based coatings with a thickness of 30 μm and 50 μm were prepared by application of 100 mg of blended PPM or copolymer with a hot press (120 °C, 8 kPa) as reported previously [9,10] (Appendix A).

### 2.5. EIS and ACET

The corrosion protection ability of the described polymer coatings was investigated by means of the accelerated cyclic electrochemical technique (ACET). All experiments were conducted by exposure of a coated area of AA2024 to a naturally aerated near-neutral solution, prepared by dissolving 0.6 mol L^−1^ (3.5 wt.%) sodium chloride in MilliQ^®^ water at room temperature (22 ± 2 °C). The pH value of the obtained solution was 6.7 ± 0.1.

The measurements were performed by using a glass cell set up as previously described [10]. A hole with a diameter of 1.1 cm (ca. 0.95 cm^2^) on the flat bottom part of the cell assures the direct contact between the coated metallic surface (working electrode) and the working solution (0.6 M NaCl). In order to avoid leakage of the working solution during the measurements, a sealing bi-adhesive layer (a2 Soluzioni Adesive, Milan, Italy) between the polymer coating and the glass cell was applied. An aqueous saturated calomel electrode (E_SCE_ = 0.242 V vs. SHE) was employed as a reference electrode, and a platinum coil was used as a counter electrode to avoid charge accumulations. The measurements were conducted on a biologic-VMP3 electrochemical workstation. The EIS analyses were conducted in the frequency range from 100 kHz to 0.01 Hz, using a sinusoidal voltage of 10 mV as amplitude at the open circuit potential (OCP).

After a preliminary EIS analysis (control EIS), the polarization–relaxation–EIS sequence was repeated at least six times, one after the other, according to the international standard ISO 17463:2014 [27]. In detail, the cathodic polarization step was performed at −2 V vs. SCE for 20 min; the relaxation process lasted 3 h; the EIS step was carried out using the same mentioned parameters. The electrochemical data were collected and analyzed by using EC-Lab software (BioLogic Science Instruments, Göttingen, Germany). Furthermore, in order to achieve a physically reasonable open circuit potential (OCP) (or, in general, to perform any electrochemical tests), permeation of electrolytes through the coating down to the underlying metal is necessary to guarantee the mandatory electric contact between the electronic and the ionic conductors (metal and solution, respectively). Thus, with homogeneous and low porosity coatings, induction times between 3 h and 24 h were required in order to establish a reliable OCP. The intrinsic self-healing mechanism of PPM copolymer coatings was also investigated by using an accelerated cyclic electrochemical technique, applying an artificial circular scratch (diameter: 52 mm, depth 30 μm) during the already mentioned induction time. The Bode impedance spectra of the coatings were quantitatively analyzed using equivalent circuits modeled as a series of resistor R(Δx) and capacitor C(Δx) elements connected in parallel, RC(Δx), as displayed in Figure 3a. The sum of the impedances of each RC(Δx) element provides the total impedance associated to a R_eq_C_eq_ parallel, with R_eq_ and C_eq_ being the equivalent resistance and capacitance, respectively.

Due to the water concentration gradient across the film, each Δx layer contributes to the overall film capacitance with its own value according to Equation (1). As proposed in the literature for thermosetting coating materials, the capacitance of the protective film changes also as a function of time according to the water uptake rate (light blue lines in Figure 3a); thus, at least a part of the impedance variation is attributed to the water permeation through the polymer film [15,16,19,20,21]. Depending on the different electrochemical behavior of each coating, the equivalent circuits used in the fitting (Figure 3b) were designed adding, in series, parallel RC equivalent elements: one for each time constant detected from the recorded spectrum. Noteworthy, all circuits include at least one series equivalent resistance R_s_, which is the sum of the electrolyte resistance and all other ohmic resistances in the system (shunt resistance, contact resistance, etc.), and a parallel resistance RC, which accounts for the coating layer. It consists of the resistance R_c_, which is representative of the porosity and degree of deterioration of the polymer coating, coupled with the coating capacitance C_c_, which is related to water adsorption (Equation (1)). An additional second parallel RC element, comprising the polarization resistance R_dl_ and the double layer capacitance C_dl_, accounts for the polarization resistance at the solution–metal interface and for the delamination and onset of corrosion at the interface [23,28], respectively. The χ-squared parameter of the fitting curves for each coating system was always less than 0.01. The exposed surfaces were finally investigated by an optical microscope (Digital Microscope, Keyence VHX-5000, Keyence, Mechelen, Belgium) in order to report on the state of the coating.

## 3. Results

### 3.1. 50 μm Thick Coatings

The impedance spectra of 50 μm thick coatings of blended PPM (5% mol/mol BBT) and a PPM copolymer with 13.4% mol/mol of octyloxy side chains recorded during eight ACET cycles are shown in Figure 4a. The coatings exhibit only two processes of different characteristic time constants, suggesting that no significant corrosion product accumulation occurred at the metal–surface interface over the entire duration of the test. Furthermore, high and stable impedance |Z|_0.01Hz_ values are recorded after each ACET cycle (Figure 4b), suggesting the preservation of the original barrier effect, especially for PPM copolymer coating, even after many aging cycles.

Coherently, for the entire duration of the ACET, the coating capacitance (Figure 5) of both systems (proportional to the water uptake) exhibits an almost stable value around 0.1 nF cm^−2^ (Figure 5). The only exception to the stable trend is the sudden increases in capacitance during the 4th and 7th cycle for PPM copolymer and blend, respectively. Those, coatings invariably restore their C_c_ to the original values in the subsequent cycle.

### 3.2. 30 μm Thick Coatings

#### 3.2.1. Behaviour under Accelerated Aging Conditions

Figure 6a shows the results of the last EIS investigation performed during the ACET tests on 30 μm thick coatings. A decrease in impedance arises over the ACET cycles, coupled with the appearance of a third time constant in the EIS spectrum after three cycles arise for the blended PPM coating. On the other hand, the PPM copolymer coating presents |Z|_0.01Hz_ above 10^8^ Ω cm^2^, even after seven ACET cycles and all the spectra are properly fitted by a two-time constant electric equivalent circuit. Interestingly, in the first cycle of the ACET test for the PPM copolymer coating, a value of 10^7^ Ω cm^2^ for |Z|_0.01Hz_ is observed. This value increases (1.5 orders of magnitude) during the second cycle, settling its value between 10^8^ and 10^9^ Ω cm^2^ for all subsequent cycles, as shown in Figure 6b. As a result, the PPM copolymer coating exhibits a value for |Z|_0.01Hz_ which is ca. one thousand-fold higher than that of the blended PPM coating.

From the fitting parameters displayed in Figure 7a,b, the coating capacitance C_c_ (a) and the related coating resistance R_c_ (b) of the blended PPM coating oscillate over the exposition time and stabilize after 5 cycles around 10^−9^ F cm^−2^ and 10^4^ Ω cm^2^, respectively. In contrast, the PPM copolymer coating presents a lower C_c_ value coupled with a higher pore resistance (notwithstanding the small decrease detected during the last cycle), both with very stable values cycle after cycle. Both parameters point towards a good stability of the coating lasting for all the duration of ACET cycles. Furthermore, the metal–coating interface of the blended PPM coating presents a slightly higher double layer capacitance and a lower resistance than the PPM copolymer coating (Figure 7c,d), evidencing a more active interface (e.g., a wider area of the metal in contact with the electrolyte).

The current densities measured during the first and the last cathodic polarization (i.e., the stress stage of ACET) for 30 µm thick coatings show an increase of about one order of magnitude for the blended PPM sample (Figure 8), which is experimental evidence of a decrease in the overall protective effect of the coating during the aging test (i.e., an increase in the extent of the underlying metal surface exposed to the solution). In contrast, the measured current density for the PPM copolymer coating decreases about four orders of magnitude over the ACET period time showing an enhanced insulating behaviour (i.e., lower current, hampered corrosion).

These chronoamperometry responses match well, not only with the aforementioned EIS monitoring, but also with the results coming from optical microscopy investigations performed on the post-treated samples (Figure 9). At the end of the ACET test, the blended coating clearly evidences at least one hole in the protective coating, responsible for the occurrence of a localized corrosion event at the surface of the exposed AA2024 (occurring during the relaxation step of each ACET cycle) and for the increased cathodic current flowing during the last polarization step (i.e., wider active surface, higher current). The defect in the polymer coating is visually identified as a black spot on the surface of the sample when irradiated with UV light at a 254 nm wavelength (Figure 9b), as a consequence of the lack of green fluorescence emission coming from the blended PPM layer all around it. The occurrence of the damage in the PPM coating is also associated with an intensification of the fluorescence emission just around the hole, a further useful diagnostic tool in identifying even small breaks in the coating (see for example the centre of the sample in Figure 9b). In contrast, the aged PPM copolymer coating does not show any hole/break in the polymer matrix, as evidenced by the lack of any black spot under 254 nm light irradiation (Figure 9a). Areas corresponding to darker small spots detected under visible light do not show any variation in the intensity of the PPM fluorescence, pointing to the temporary occurrence of localized corrosion events in correspondence to small meta-stable holes/cracks in the polymer film that were healed during the ACET cycle, possibly during the 3 h long relaxation stages by exploiting the heat flow generated from the aluminium corrosion (see Discussion section).

#### 3.2.2. Self-Healing Assessment on PPM Copolymer Coating after Mechanical Damage

In view of the interesting behavior shown during ACET cycles, the self-healing ability of the PPM copolymer coating was further investigated. To provide additional proof of the supposed self-healing ability of PPM copolymer, a controlled defect in the coating was introduced. To do so, a AA2024 specimen covered with a pristine PPM copolymer coating was exposed to synthetic sea water and, after a proper induction time, an artificial scratch was applied to the coating in order to establish a direct physical and electric contact between the solution and the metal surface. As reported in Figure 10a, during the initial induction time (ca. 2.8 h), a perfect insulation behavior of the coating was observed as confirmed by the establishment of an OCP value without physical meaning (ca. −13 V vs. SCE), indicating that the specimen was perfectly isolated from the reference electrode.

After the artificial scratch was applied, a sudden drop of the OCP at around −0.5 V and −0.7 V (vs. SCE) was detected. These values are compatible with naked aluminum alloys in chloride solution.

During the cathodic polarization step (Figure 10b), the current density decreases by about four orders of magnitude in the first few minutes. This drop means that the coating quickly established a more protecting feature, attributable to the self-healing of the artificial crack. This process ends and stabilizes during the subsequent 3 h relaxation period (Figure 10c) in which the system is let free to set at its rest potential. The newly established insulating condition (resulting from self-healing of the original crack) lasts for an additional 24 h, before a final significant loss of the protective feature (|Z|_0.01Hz_ drops from 10^8^ to 10^6^ Ω cm^2^) under accelerated aging conditions occurs. The Bode modulus plots, depicting the system during the last aging cycles, are collected in Appendix A.

## 4. Discussion

### 4.1. Effect of the Formulation and the Thickness of PPM-Based Coatings

The rate of corrosion in a coated system depends on water uptake, ion diffusion and conduction through the protective layer, but also on the nature of the corrosion reaction. In the case of aluminum alloy in presence of chloride ions, the corrosion proceeds via an autocatalytic mechanism [29]. The coatings presented in this article often show a stable Bode impedance (e.g., |Z|_0.01Hz_ > 10^7^ Ω cm^2^) without the formation of visible corrosion spots. In these cases, a second time constant element, which fits the low frequency region of the Bode impedance spectra, appears. In the absence of visible corrosion spots and recording low and stable current densities, the second time constant element could be indirectly attributed to the diffusion-controlled phenomena related to the mobility of the ions through the pores [30]. Accordingly, this phenomenon is predominant when the potential is oscillated at low frequencies affecting, importantly, the impedance of the system. Notably, the dimension and the shape of pores also affect the shape of the impedance spectrum [31]. Thus, the second time constant element in our systems is not ascribed to the appearance of a new interface due to the formation of corrosion products but can be ascribed to the ionic diffusion phenomena that are still not enough to trigger the corrosion reactions.

Both blended PPM and copolymer coatings of 50 μm thickness show a similar behavior in terms of corrosion protection, with a good physical barrier property (assessed through the low frequency impedance, |Z|_0.01Hz_) and a limited water uptake over time (assessed through the coating capacitance, C_c_ [15,16,19]). Notwithstanding the forced cathodic polarizations, the Bode modulus plots for the two 50 µm-thick coatings maintained the shape of a two-constant time system even after eight ACET cycles, with constant values of the low frequency impedance about 10^7^ Ω cm^2^. Both features point to a uniform defect-free and adherent insulating layer, endowed with a slow tendency to uptake water, that efficiently preserves AA2024 from wet corrosion. Interestingly, during the ACET experiments, the coating capacitance of the blended PPM coating and the PPM copolymer coating exhibits a maximum at the 7th and 4th cycle, respectively. The evidence that these peaks occurred after several cathodic polarizations suggests that some porosities or defects are created over the accelerated ageing of the coating, but that they are somehow recovered in the following step. This uncommon behavior can be ascribed to the self-healing activity already tentatively reported for PPM-based coatings [9].

ACET tests on 30 μm thick coatings (Figure 6a,b) revealed a neatly poorer performance of the blended PPM coating when it is processed at this thinner thickness. Indeed, as a consequence of the aforementioned porosity, a higher amount of water can pass through the blend coating reaching the underlying metal surface where, during the cathodic polarization stage, the electron-assisted reduction of water enhances defects of the coating, leading to a faster coating failure (e.g., delamination, degradation, corrosion). This is confirmed by the decrease in |Z|_0.01Hz_, cycle after cycle (Figure 6a), and by the increased current density recorded during the last polarization stage of ACET (Figure 8), both data being coherent with an increase in the exposed area of the aluminum alloy.

During the first ACET cycle on the PPM copolymer coating the impedance module (ca. 10^7^ Ω cm^2^) and the current density measured during the cathodic polarization (ca. 10 mA cm^−2^) imply the presence of enough porosity or defects in the coating to trigger the electrochemically assisted aging that should lead to an incumbent coating deterioration/failure over the ageing cycles. However, after the first ACET cycle, the systematic increase in |Z|_0.01Hz_ (Figure 6), the trend of the values of the equivalent circuit elements (Figure 7) and the decrease in the current density (down to few μA cm^−2^; Figure 8) reveal a possible intrinsic self-healing of coating porosities, which enhances the corrosion protection ability of the coatings prepared with the PPM copolymer. Due to the unconventional behavior of this coating, a further assessment of the self-healing was carried out using the ACET on an artificially damaged surface. The experiment revealed that the coating porosity and the surface damage were quickly healed during the first cathodic polarization (Figure 10b). The high insulation of the metallic substrate was also confirmed by the potential recorded in the following relaxation step. In addition, over the relaxation step (Figure 10c), new defects through the coating were established but immediately self-healed, restoring the high insulation behavior of the AA2024 for another 24 h. The value of |Z|_0.01Hz_ in the following ACET cycles confirmed a corrosion protection ability of the self-healed PPM copolymer coating under accelerated ageing, even after mechanical damages.

### 4.2. Polymer Rearrangement under Localized Corrosion

The evidence that self-healing takes place during both cathodic polarization and the relaxation step suggests that the self-healing mechanism can be triggered by thermal shocks. These could be tentatively ascribed to the Joule effect during the cathodic polarization, and to corrosion reactions of aluminum during the relaxation step. According to the literature, aluminum alloys are significantly corroded by chloride ions typically in two highly exothermic steps [32]. At the beginning, Cl^−^ attacks the aluminum, forming AlCl_3_. Immediately afterwards, AlCl_3_ strongly reacts with water, resulting in the formation of aluminum hydroxide [29,33] and HCl, which regenerates the autocatalytic attack.

Furthermore, the pure and reversible thermoplastic behavior of the PPM copolymer, and the amorphous nature of PPM and its derivatives [10,12], suggest that after thermal shocks, a flow of the polymer matrix could locally arise. Notably, this combination of materials properties cannot be addressed by thermosetting polymers (e.g., epoxy or alkyd resins) or by other coatings containing substances sensible to corrosion side reactions (such as oxidation and hydrolysis) [34].

The unconventional increase in |Z|_0.01Hz_ and the almost constant values of the electrochemical fitting parameters over the exposure time could be tentatively explained by combining the material properties of the PPM copolymer with the exothermic steps involved in the AA2024 corrosion or eventually the Joule effect that could arise over the cathodic polarization. In absence of other chemical reactions of PPM [9,10,11,12], it is supposed that the copolymer undergoes intrinsic self-healing, probably by flowing and thus closing the pores in the matrix that induces (or that are formed after) localized corrosion events.

## 5. Conclusions

Using the ACET, the corrosion protection properties of 50 μm and 30 μm thick poly(phenylene methylene)-based coatings were assessed under “accelerated aging”. The investigation specifically aimed at comparing the features of coatings formed by a blended PPM, containing an external plasticizer (BBP), versus a PPM copolymer, containing an internal plasticizer (octyloxy-PPM copolymer).

The investigation revealed that in 50 μm thick coatings stable |Z|_0.01Hz_ about 10^8^ Ω cm^2^ are observed for both substrates. In contrast, a decrease in coating thickness to 30 μm for blended PPM coatings leads to a significant decrease in corrosion protection capability, as reflected by a decrease in |Z|_0.01Hz_ after two ageing cycles to 10^6^ Ω cm^2^. On the other hand, after thickness optimization to 30 μm, PPM copolymer coating shows an unexpected behavior during the accelerated test with an increase in |Z|_0.01Hz_ from 10^7^ Ω cm^2^ to values higher than 10^8^ Ω cm^2^. The trend over time of the experimental electrochemical parameters of the PPM copolymer can be explained by a healing of pre-existing (or even newly formed) pores in the polymer layer. The healing of the surface was also confirmed by optical microscopy images taken under 254 nm light irradiation capable of stimulating PPM fluorescence emission.

Furthermore, the ACET performed on an artificially damaged substrate revealed that self-healing (prerogative of PPM copolymer only) takes place both during the cathodic polarization and during the relaxation time as evidenced by the decrease in the current density, and the establishment of an almost perfect electrical insulation of the working electrode (very low, physically meaningful OCP values). These observations suggest that the phenomenon can be ascribed to exothermic heat fluxes produced during both processes. The thermoplastic behavior of the PPM copolymer was identified as the predominant material feature able to justify the observed self-healing behavior. The local flow of the polymer was triggered by the heat released by the reactions involved in the localized corrosion of aluminum alloy in the presence of chloride ions.

In conclusion, the encouraging corrosion protection ability, the thermoplastic behavior and the self-healing ability of PPM derivatives under corrosion attack led us to believe that this type of materials (especially when prepared by copolymerization with a suitable co-monomer) represents a promising thermoplastic alternative to the thermosetting resins commonly used for metallic corrosion protection.

## Figures and Tables

**Figure 1 polymers-14-03457-f001:**
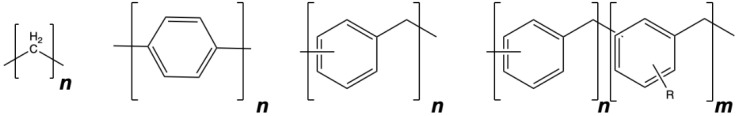
Chemical structure (from left) of polyethylene, poly(*p*-phenylene), poly(phenylene methylene), and poly(phenylene methylene) containing side chains (R = 4-octyloxy).

**Figure 2 polymers-14-03457-f002:**
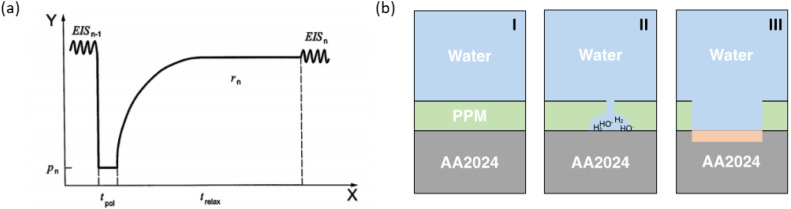
(**a**) Scheme of the ACET test in which X is the experimental time, Y is the electric potential, p_n_ represents the cathodic potential applied during the polarization step, t_pot_ and t_relax_ stands for the time of the polarization and relaxation, respectively. (**b**) Schematic representation of coating life, (I) represents a healthy coating, (II) delamination and/or porosity formation with subsequent blistering, (III) corrosion propagation.

**Figure 3 polymers-14-03457-f003:**
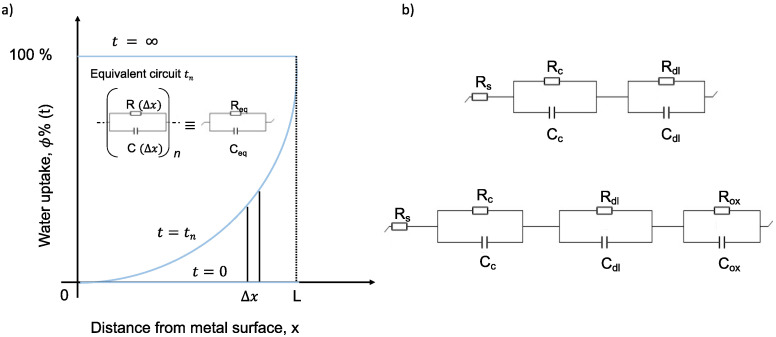
(**a**) Schematic representation of a water uptake profile in a coated system of thickness L and equivalent electric circuit following the continuous model (CM); (**b**) equivalent circuits (with two and three characteristic time constants) used to model EIS and ACET data of the analyzed coated samples.

**Figure 4 polymers-14-03457-f004:**
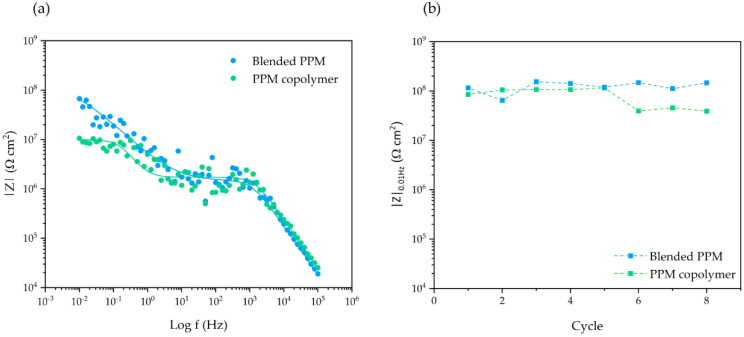
Evolution of Bode modulus plot in ACET experiments during the last cycle (**a**); the fitted data are represented by lines while the experimental values are represented by circles. Evolution of |Z|_0.01Hz_ over the whole ACET cycles (**b**); dotted lines represent a guide for eye only. In both graphs: 50 μm coating of PPM copolymer (green) and of blended PPM (blue).

**Figure 5 polymers-14-03457-f005:**
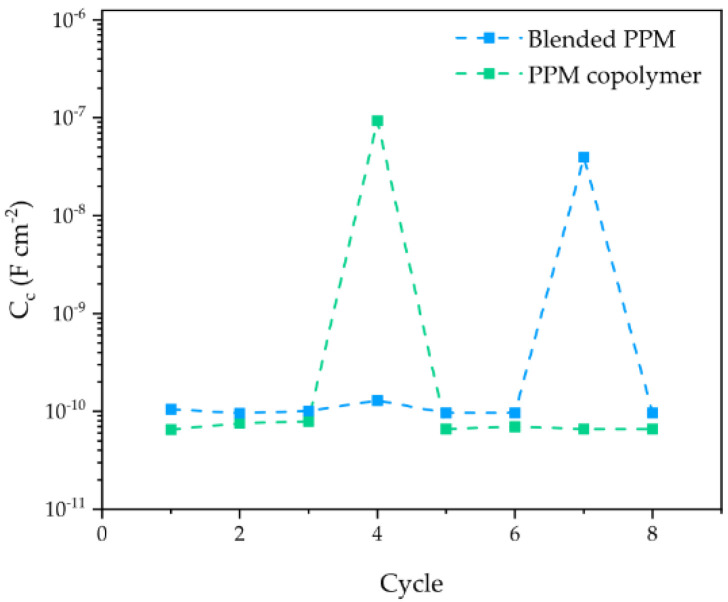
Evolution of the coating capacitance Cc for a 50 μm PPM copolymer coating (green) and a 50 μm coating of blended PPM (blue) over ACET cycles. Values are obtained by the fitted EIS spectra carried out at OCP. Dotted lines represent a guide for eye only.

**Figure 6 polymers-14-03457-f006:**
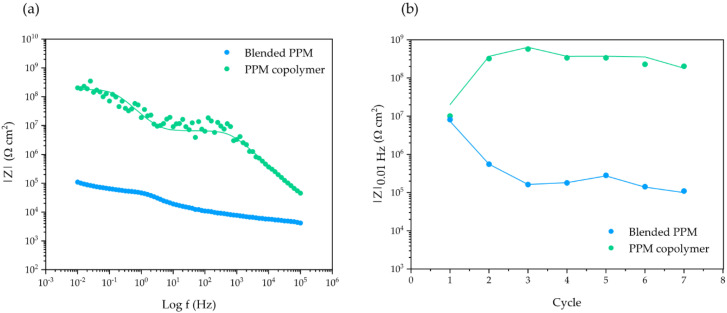
Evolution of the Bode modulus plot in ACET experiments during the last cycle (**a**); the fitted data are represented by lines while the experimental values are represented by circles. Evolution of |Z|_0.01Hz_ over the whole ACET cycles (**b**); dotted lines represent a guide for eye only. In both graphs: 30 μm coating of PPM copolymer (green) and of blended PPM (blue).

**Figure 7 polymers-14-03457-f007:**
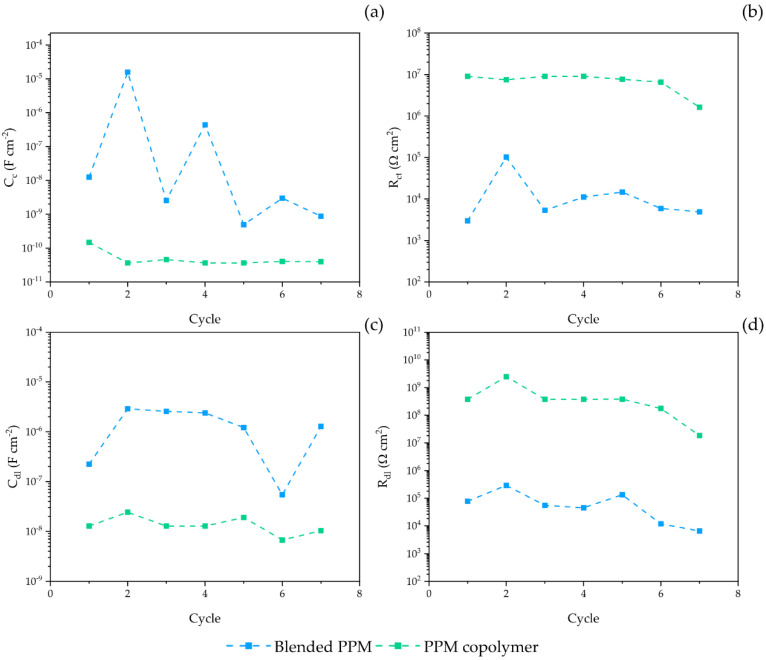
Evolution of the coating capacitance C_c_ (**a**), pore resistance (R_c_), polarization resistance (R_dl_) and double layer capacitance (C_dl_) during the ACET test of a 30 μm PPM copolymer coating (green) and a 30 μm coating of blended PPM (blue). Dotted lines represent a guide for eye only.

**Figure 8 polymers-14-03457-f008:**
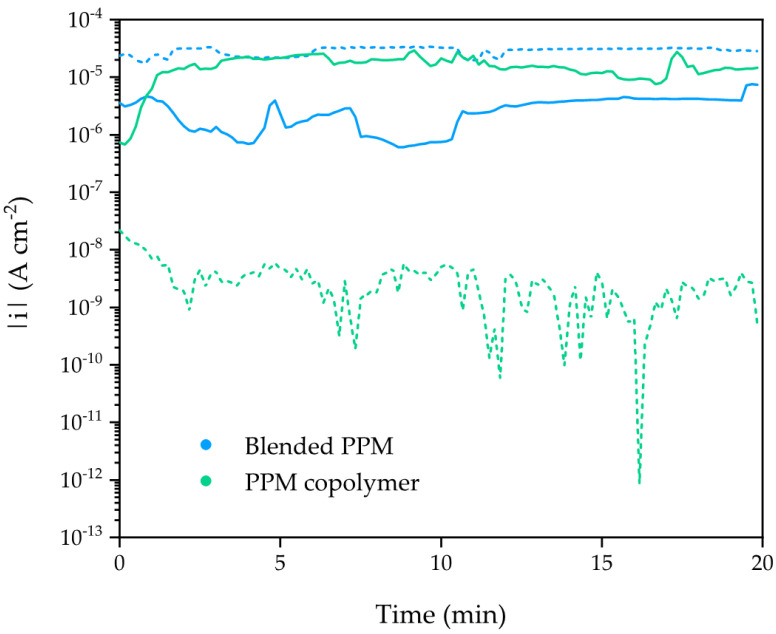
Current density (in logarithmic scale) of 30 μm PPM-based coatings recorded during the first (solid line) and the last (dotted line) cathodic polarization at −2 V vs. SCE. PPM copolymer coating is shown in green and the blended PPM in blue.

**Figure 9 polymers-14-03457-f009:**
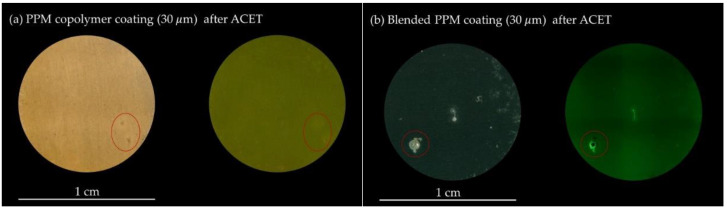
Optical microscopy images of the surface of (**a**) 30 μm PPM copolymer and (**b**) 30 μm blended PPM coating after exposition to the ACET under visible light (**left**) and under UV light at 254 nm (**right**). Macroscopic localized corrosion of the AA2024 substrate is marked with red circles.

**Figure 10 polymers-14-03457-f010:**
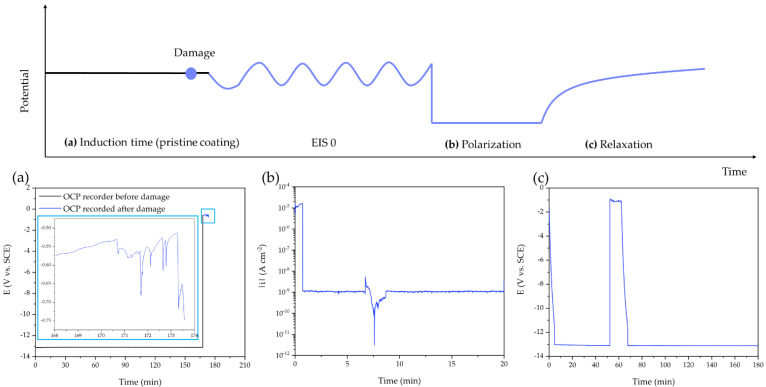
Evolution over time of some key electrochemical parameters (OCP (**a**,**c**) and current density (**b**)) of a specimen with PPM copolymer coating before (black line) and after an artificial damage (blue line).

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
