# Peer review of "Smart Anticorrosion Coatings Based on Poly(phenylene methylene): An Assessment of the Intrinsic Self-Healing Behavior of the Copolymer"

_polymers, 2022, doi:10.3390/polym14173457_

Round 1

Reviewer 1 Report

In this well-written manuscript, D’Elia et al. processed and characterized PPM based smart thermoplastic anticorrosion coatings and studied its intrinsic self-healing behavior. The subject matter could be interesting for the general readership of the journal. I am in favor of publishing this manuscript containing interesting data and relevant analysis. My comments are given below for a minor revision:

- “… however, bismuth (III) trifluoromethanesulfunate (0.32 g, 0.39 mmol) was employed as catalyst instead of tin tetrachloride (see Supplementary Material file S.M.)” – Why was this change made? Please correct the typo in SM.

- Self-healing assessment (further characterization related to Fig. 8): Fig 8 is not conclusive. Please provide SEM images of the coating surface and compare. 

Author Response

Me and other co-authors are pleasant to know that the aim of the Journal is respect within our manuscript.  The points listed in the report dully implemented in the new version of the manuscript.

Reviewer wrote:

however, bismuth (III) trifluoromethanesulfunate (0.32 g, 0.39 mmol) was employed as catalyst instead of tin tetrachloride (see Supplementary Material file S.M.)” – Why was this change made? Please correct the typo in SM.”

Answer: 

We are in agreement with the Reviewer that a better explanation on why we used Bismuth trifluoro methanesulfunate as catalyst is necessary. Lewis acids are reported to be good catalyst for the polymerization of benzyl chloride (I.e. 11,12). Thus, we decided to respect the operative condition for the synthesis of PPM copolymers reported in our previous work [10] just replacing SnCl4 with another Lewis acid as Bismuth trifluoro methansulfunate which is a less hazardous chemical. The polymer obtained didn’t show major differences with the counterpart reported in literature (similar thermal properties and molar masses) so we decided to don’t stress this point to don’t shift the focus of the article.

Reviewer wrote:

“Self-healing assessment (further characterization related to Fig. 8): Fig 8 is not conclusive. Please provide SEM images of the coating surface and compare”

Answer: 

We agree the reviewer that SEM pictures could help in a better understanding of the coating properties. Indeed, we already tried to use this technique to investigate self-healing of our coatings. However, the strongly insulating properties of this coating polymer do not allow any SEM evaluation without the application of a conductive top layer (i.e., metallization) and, even in this condition, the heat generated by the electron flux over the analyzed spots during the electronic scanning invariably gradated the surface. This unfortunately precludes a clear evaluation of the coating surface with this technique.  For these reasons, we decided to further confirm the hypothesis argued from  impedance analysis (Figure 7) and current density curves (Fig 8) by exploiting optical microscopy to track the photoluminescence of the polymer (Figure 9). Indeed, exploiting the photoluminescence activity of PPM is possible to detect  defects and cracks (caused and/or induced by corrosion spots or meta-stable pits) along the coating which cause light diffractions . In particular, when a coating  is exposed to UV light, any black spots observed reveal a non-protected metal area that was subjected to the  attack by corrosion species (i.e., coating failed during the previous corrosion test). On the contrary, if no black spots are detected the optically active polymers is still protecting the metallic surface. As shown in Figure 9, while the PPM blend showed a black spot (highlighted by circle), the PPM copolymer does not. The latter only shows few darker zones under Vis light (partially corroded AA areas) but they are perfectly uniform under UV light. Combining this observation (PPM copolymer is intact) with those argued by electrochemical analyses we can infer that the coating was self-healed so covering the underlying metal spots.

Reviewer 2 Report

This paper was well prepared and organized from the point that it showed the effect of self-healing of PPM to protect corrosion of the substrate. But some part of the manuscript needs to be revised as follows;

1. Abstract needs some information on EIS results.

2. Experimental

  1) The manuscript used two kinds of specimen - blended PPM and copolymer PPM. Where is the reference specimen? Do you want to compare two PPMs? If it is, please change the title of the manuscript to show the purpose clearly?

  2) I recommend the simple name of the specimen as like 'Blended PPM' and 'Copolymer PPM' etc.

  3) '2.5 EIS and ACET' is too long and thus it should be briefly rewritten.

3. Results

  1) Fig. 3 and Fig. 5 and Fig. 6 and Fig. 7 and Fig. 11 need the legend in the figures. 

  2) Fig. 11; Which is the specimen used? 'blended or copolymer' Blue line was the 3rd EIS result and the properties was degraded. Where is the comparative specimen? How can you show the self-healing effect?

4. Conclusions

  1) Just 1 paragraph. Please rewrite the conclusions using several paragraphs.

  2) The manuscript concludes the self healing effect using two specimen. But I think the manuscript needs the reference specimen because the protection effect could be achieved even though the coating used didn't have 'blended or copolymer'. I guess Fig. 10 and Fig. 11 were the results about the self-healing effect. Is it right? How could you confirm the self healing effect? 

Author Response

The others co-authors and me are grateful to know that the Reviewer appreciated our work and we want to thank him for the advice and for the useful suggestion provided in the report. Here below a detailed description of the actions done on the revised manuscript.

Reviewer wrote:

Abstract needs some information on EIS results

Answer: 

We gladly modified the abstract according to the advisees. 

Reviewer wrote:

“The manuscript used two kinds of specimen - blended PPM and copolymer PPM. Where is the reference specimen? Do you want to compare two PPMs? If it is, please change the title of the manuscript to show the purpose clearly?

and:

"The manuscript concludes the self healing effect using two specimen. But I think the manuscript needs the reference specimen because the protection effect could be achieved even though the coating used didn't have 'blended or copolymer'. I guess Fig. 10 and Fig. 11 were the results about the self-healing effect. Is it right? How could you confirm the self healing effect?"

Answer: 

The revised manuscript was modified (starting for the title and the abstract, too) to better clarify that the topic of this work is a detailed comparison between PPM blend (our reference specimen) and PPM copolymer. It is important to underline that a comparison with pure PPM is not possible due to the extreme coating brittleness (Figure 1, here below). As consequence, PPM has to be blended with plasticizer to be processed into anticorrosion coatings. Thus, for example, we modified the text in the introduction to better clarify this crucial point, that is that the PPM blend must be considered as the reference  specimen. 

see attached Figure 1 Pure PPM coating (not blended neither copolymerized).

Reviewer wrote:

I recommend the simple name of the specimen as like 'Blended PPM' and 'Copolymer PPM' etc

Answer:

We perfectly agree with his/her comment. We have adopted the “shorter” advised nomenclature (PPM blend vs. PPM copolymer) to distinguish better the PPM copolymer and the blend.  

Reviewer wrote:

“ ’2.5 EIS and ACET' is too long and thus it should be briefly rewritten.

Answer:

The EIS and ACET paragraph in the experimental section was modified according to the Reviewer’s comments and reduced about 1/4 in order to make it shorter and less redundant.  

Reviewer wrote:

Fig. 3 and Fig. 5 and Fig. 6 and Fig. 7 and Fig. 11 need the legend in the figures.

Answer:

We thank the Reviewer. Besides the description provided in each caption, we’ve added the legend in the Figures.

Reviewer wrote:

Fig. 11; Which is the specimen used? 'blended or copolymer' Blue line was the 3rd EIS result and the properties was degraded. Where is the comparative specimen? How can you show the self-healing effect?

Answer:

Answering the question about how we can proof self-healing of PPM copolymer, actually, none of the provided experimental proof, if taken individually, is able to incontrovertibly confirm the advent of self-healing events (i.e., enclosure of microporosity, very small holes, etc). On the contrary, looking as a whole to the clues provided by the many techniques here employed (often complementary to each other), a sufficiently clear picture can be obtained that points to PPM copolymer (30 µm in thickness) being a system that increases its barrier properties over time (even under accelerated stress conditions) and that does not show evident holes/cracks at the end of the tests, contrary to the reference specimen (i.e., blend PPM).

Reviewer wrote:

“Just 1 paragraph. Please rewrite the conclusions using several paragraphs

Answer:

Accordingly, we have rewritten the conclusion section.

Reviewer 3 Report

According to the manuscript with title: “Smart thermoplastic anticorrosion coatings based on poly(phe- 2 nylene methylene): assessment of the intrinsic self-healing be- 3 havior". The submitted work is introducing a new valuable and interesting idea and the given results confirm the idea. This work is suitable for publication in the Journal. I suggest the acceptance after some major corrections as follows;

1.     Abstract section need to rewrite in correct sequence with more explanation

2.     Add some critical numbers in abstract section

3.     I don’t understand figure 7

4.     What obtain figure 8 b?

5.     Reformulate the aim of the work in introduction

6.     Add previous published work with comparison to clear the novelty of your work

7.     Give some results with numbers in conclusion

8.     Add more explanation to experimental work

9.     Correct typographical errors.

Author Response

The others co-authors and me are grateful to know that the Reviewer appreciated our work and we want to thank him for the advice. The suggested modifications were properly integrated in the revised manuscript. Here below a detailed description.

Reviewer wrote:

Abstract section need to rewrite in correct sequence with more explanation"

  Add some critical numbers in abstract section

Answer:

We gladly modified the abstract according to the advisees. 

Reviewer wrote:

I don’t understand figure 7

   What obtain figure 8 b? 

  Add more explanation to experimental work"

Answer

The text was integrated in order to make the Figures and the related discussion clearer and more understandable to readers. Moreover, modifications to the Experimental sections are also provided.

Reviewer wrote:

“ Reformulate the aim of the work in introduction

   Add previous published work with comparison to clear the novelty of your work"

Answer

Thus, we modified the introduction to better clarify the novelty of this work. It consists in a comparison of the corrosion protection properties of PPM-based coatings with external (blend PPM) or internal plasticizers (PPM copolymer) via  i) a modification of coating thickness and ii) a detailed investigation of the corrosion protection features over time based on the accelerated electrochemical techniques (ACET). Firstly, through the coating thickness optimization, the best strategy of coating preparation is assessed (i.e., copolymerization). Moreover, the self-healing of the PPM copolymer coating is proven by synergistically combining electrochemical methods (systematically discussing in deep details many key parameters) and fluorescence investigation using optical microscopy.    

Reviewer wrote:

Give some results with numbers in conclusion "

Answer

Conclusions was properly improved, as suggested.

Round 2

Reviewer 2 Report

Additional work about the self-healing effect may be needed.